# Comparative analysis of silver-nanoparticles and whey-encapsulated particles from olive leaf water extracts: Characteristics and biological activity

Hanem M. M. Mansour[1]*, Mohamed G. Shehata[1,2], Eman M. Abdo[3]*, Mona Mohamad Sharaf[4], El-sayed E. Hafez[5], Amira M. Galal Darwish[1,6]

1 Food Technology Department, Arid Lands Cultivation Research Institute (ALCRI), City of Scientific Research and Technological Applications (SRTA-City), Alexandria, Egypt, 2 Food Research Section, R&D Division, Abu Dhabi Agriculture and Food Safety Authority (ADAFSA), Abu Dhabi, United Arab Emirates, 3 Food Science Department, Faculty of Agriculture (Saba Basha), Alexandria University, Alexandria, Egypt, 4 Protein Research Department, Genetic Engineering and Biotechnology Research Institute, City of Scientific Research and Technological Applications (SRTA-City), Alexandria, Egypt, 5 Plant Protection and Bio-Molecular Diagnosis Department, Arid Lands Cultivation Research Institute, City of Scientific Research and Technological Applications (SRTA-City), Alexandria, Egypt, 6 Food Industry Technology Program, Faculty of Industrial and Energy Technology, Borg Al Arab Technological University (BATU), Alexandria, Egypt

* hanm_m123@yahoo.com (HMMM); Eman-abdo@alexu.edu.eg (EMA)

**Data Availability Statement:** All relevant data are within the paper and its Supporting information files.

## Abstract

Nanotechnology applications have been employed to improve the stability of bioactive components and drug delivery. Natural-based extracts, especially olive leaf extracts, have been associated with the green economy not only as recycled agri-waste but also in the prevention and treatment of various non-communicable diseases (NCDs). The aim of this work was to provide a comparison between the characteristics, biological activity, and gene expression of water extract of olive leaves (OLE), green synthesized OLE silver nanoparticles (OL/Ag-NPs), and OLE whey protein capsules (OL/WPNs) of the two olive varieties, Tofahy and Shemlali. The particles were characterized by dynamic light scattering, scanning electron microscope (SEM), and Fourier transform infrared. The bioactive compounds of the preparations were evaluated for their antioxidant activity and anticancer effect on HCT-116 colorectal cells as well as for their regulatory effects on cytochrome C oxidase (Cox1) and tumor necrosis factor α (TNF-α) genes. (OL/Ag-NPs) were found to be smaller than (OL/WPNs) with sizes of (37.46±1.85 and 44.86±1.62 nm) and (227.20±2.43 and 553.02 ±3.60 nm) for Tofahy and Shemlali, respectively. SEM showed that Shemlali (OL/Ag-NPs) had the least aggregation due to their highest ξ-potential (-31.76 ± 0.87 mV). The preparations were relatively nontoxic to Vero cells ($IC_{50}$ = 151.94–789.25 μg/mL), while they were cytotoxic to HCT-116 colorectal cells ($IC_{50}$ = 77.54–320.64 μg/mL). Shemlali and Tofahy OLE and Tofahy OL/Ag-NPs had a higher selectivity index (2.97–7.08 μg/mL) than doxorubicin (2.36 μg/mL), indicating promising anticancer activity. Moreover, Shemlali preparations regulated the expression of Cox1 (up-regulation) and TNF-α (down-regulation) on HCT-116 cells, revealing their efficiency in suppressing the expression of genes that promote cancer cell proliferation. (OL/Ag-NPs) from Tofahy and Shemlali were found to be

**Funding:** The author(s) received no specific funding for this work.

more stable, effective, and safe than (OL/WPNs). Consequently, OL/Ag-NPs, especially Tofahy, are the best and safest nanoscale particles that can be safely used in food and pharmaceutical applications.

## 1. Introduction

Olive (*Olea europaea* L.) leaves represent about 10% of the olive oil industry's by-products [1], with more phenols than other olive wastes. Oleuropein is the most abundant in the leaves (60–90 mg/g dry weight), while luteolin-7-glycoside and hydroxytyrosol have also been detected in high concentrations [2]. Consequently, leaves showed anti-diabetic, anti-inflammatory, antimicrobial, anticancer [1], antiproliferative, and apoptotic activities [3]. However, phenols from olive leaves have an unpleasant taste [4], low bioavailability, and low stability [5], which limits their potential applications [4]. Therefore, micro and nanotechnologies, such as silver nanotechnology and encapsulation, are used to overcome these limitations. Nanoparticles (NPs) can be engineered to encapsulate and protect phytochemicals from degradation, improve their solubility, and increase their bioavailability. In addition, NPs can be functionalized with targeted ligands to selectively deliver drugs to specific cells or tissues to reduce off-target effects and improve therapeutic efficacy. NPs can also be designed to release olive phytochemicals in a controlled manner, enabling sustained drug release over a long time [6].

Green synthesis of silver NPs (AgNPs) is an innovative, eco-friendly, and sustainable alternative to traditional chemical and physical synthesis methods. This approach utilizes plant extracts due to their high content of phenolic acids, flavonoids, and amides that can reduce and cap silver ions to form stable silver NPs [7–9], thus providing an environmentally friendly, less toxic, and cost-efficient NPs [9, 10]. These green-synthesized NPs have shown promising applications in antibacterial [7, 8, 11] and anticancer therapies [12–14], highlighting their superiority over NPs produced by conventional methods. The Ag component of Ag-Silicalite-1 zeolite nanomaterial is an antimicrobial agent that can disrupt the cell membrane of *Candida auris*, leading to cell death and its broad-spectrum antimicrobial activity [15]. AgNPs have been shown to damage the ultrastructure of cancer cells [16] and cause the formation of reactive oxygen species (ROS), apoptosis [17], necrosis, and DNA damage [18]. They can also modulate various signaling pathways in the cell [19]. Thus, AgNPs offer a targeted approach that prevents undesirable side effects, has good pharmacokinetics and precise targeting, and reduces multidrug resistance [15, 18].

Olive leaf extract has demonstrated its phyto-reducing and phyto-capping ability in the biological synthesis of silver nanoparticles (Ag-NPs) [20]. The functional groups of oleuropein [21], flavonoids, terpenoids, carboxylic acids, quinones, aldehydes, ketones, and amides [22] can reduce $Ag^+$ to $Ag^0$ ions, which cluster together and subsequently become NPs. The resulting NPs showed higher antioxidant stability during storage and food processing [23–25] and offered promising effects in targeted drug therapy as antibacterial, antifungal, antiviral, and anticancer agents [26].

Encapsulating the bioactive components of olive leaves has been also successfully fulfilled using various polymeric matrices (alginate, whey, inulin, and maltodextrin) by different strategies such as freeze-drying, spray-drying, nano-emulsions, and double-emulsions [25]. The protein biopolymers, such as whey proteins, are robust wall materials. The functional groups of whey effectively bind to a variety of bioactive compounds and restrict their release until they reach the target site. Milk whey peptides have been shown to possess antihypertensive,

antiviral, anticancer, and antioxidant activities [27]. Lactoferrin, one of the components of milk whey, played a role in repairing damaged genetic material or redirecting it to apoptosis [28], as well as its explicit anticancer effect [29]. In addition, whey proteins enhance other functional properties such as foaming, emulsification, digestibility, and formation of bioactive peptides during digestion [30].

Accordingly, silver nanotechnology and encapsulation could be effectively used to enhance the bioactivity of olive leaf phenolics as a potential anticancer agent. The previous studies focused on the characterization of the prepared nano-silver and encapsulated NPs from olive leaves and their potential anticancer effects, but which particle could successfully and effectively deliver the most phenols to have a potent anticancer effect? To our knowledge, there are no previous studies that have compared the effects of nano-silver and encapsulation technologies on the properties and safety of olive leaf particles and their anti-colorectal activity through the regulation of cytochrome C oxidase (Cox1) and tumor necrosis factor-α (TNF- α) expression. Therefore, the aim of the present study was to compare the bioactive components and antioxidant activity of olive leaf water extract (OLE), green synthesized silver nanoparticles of OLE (OL/Ag-NPs), and OLE whey protein capsules (OL/WPNs) from two olive varieties, Tofahy and Shemlali. Furthermore, we investigated the cytotoxic effect of the water extracts, OL/Ag-NPs, and OL/WPNs on the normal (Vero) cell line and the anticancer effect on the colorectal (HCT116) cell line and their regulatory effect on the expression of COX and TNF-α.

## 2. Materials and methods

### 2.1. Materials

Olive leaves of two cultivars, Tofahy and Shemlali, were collected from olive trees of the same age (6 years) grown under the same environmental and agronomic conditions at the City of Scientific Research and Technological Applications (SRTA-City), New Burg El-Arab, Alexandria, Egypt.

Ethanol, glacial acetic acid, sodium carbonate, aluminum chloride, sodium chloride, and sodium hydroxide were purchased from Aljomhoria, Alexandria, Egypt. ABTS- (2, 2'-azinobis (3-ethylbenzothiazoline-6-sulfonic acid), DPPH- (2,2-diphenyl-1-picrylhydrazyl), Folin-Ciocalteu reagent, sodium azide, MTT (3-(4,5-dimethylthiazol-2-yl)-2,5-diphenyltetrazolium bromide), RPMI (Roswell Park Memorial Institute 1640) medium, DMSO (dimethyl sulfoxide), PSE (phosphate-buffered saline), and FBS (fetal bovine serum) were purchased from Merck, Darmstadt, Germany.

### 2.2 Sample preparation

The leaves were carefully washed to remove the unwanted substances, blanched at 90˚C for 2 min, and cooled directly with cold water at 15˚C. After removing the excess water with an absorbent paper, the leaves were dried in an oven (Wt-binder, Bohemia, NY, USA) at 45˚C for 3 days. Then, the ground dried leaves were stored at -80˚C for further use [31].

### 2.3 Preparation of olive leaves extracts (OLE)

Olive leaves of the two cultivars were mixed separately with hot water at 100˚C (1:50 w:v) for 10 min before the mixture was stirred at room temperature (23˚C) for 3 h. Subsequently, each sample was centrifuged at 3000 rpm/20˚C for 10 min; the supernatants were filtered and lyophilized using a vacuum freeze dryer (FDE 0350, Humanlab Inc., Bucheon-si, Gyeonggi-do, Korea) [32]. The freeze-dried extracts were stored at -80˚C for further use.

## 2.4 Green synthesis of silver nanoparticles (OL/Ag-NPs)

Olive leaf extract of each cultivar was mixed with silver nitrate ($AgNO_3$) solution (1 mM) in a dark flask to avoid $AgNO_3$ photoactivation. The mixture was shaken at 250 rpm until the $AgNO_3$ was reduced to $Ag^+$ ions. Complete reduction of $AgNO_3$ was achieved when the mixture turned from a colorless to a colloidal brown solution. Ag-NPs formation was confirmed by measuring the absorbance of the mixture at 540 nm—the process was stopped when the absorbance decreased. The solution was then centrifuged at 12000 rpm for 30 min. The pellets were collected, washed three times with d.$H_2O$, dried at 50˚C/24 h, and re-dissolved in d.$H_2O$ [33].

## 2.5 Olive leaves extracts whey protein capsules (OL/WPNs)

WPI was dissolved in 10 mM NaCl solution (3% w:v) by stirring at 500 rpm at room temperature for 2 h in the presence of sodium azide (50 mg/L) to prevent microbial growth. The solution was stored at 4˚C for 12 h. Then, the solution was filtered through a polyvinyl difluoride "PVDF" syringe filter (0.45 μm) before heating at 60˚C for 30 min. Then, 0.045 g of extract powder was added to the solution to obtain a mass ratio of 1:20. The pH was adjusted to 9.0 with 2 M NaOH to obtain smaller particles (based on preliminary experiments). Then, the solution was stirred at 500 rpm, and the ethanol was added simultaneously at a rate of 1 mL $min^{-1}$ until the solution became turbid, about 3.3 mL ethanol/mL protein solution. The particle suspension was centrifuged at 18,000 rpm for 10 min. The resulting particles were then vacuum dried at 60˚C and stored at −80˚C until analysis [34].

## 2.6 Particles properties and physical characteristics

Physical properties of particles were characterized by measuring the particle size, ζ-potential, and polydispersity index (PDI) based on the dynamic light scattering (DLS) technique using the Zetasizer Nano series (Malvern Instruments, Malvern Nono-Zs ZEN3600, UK) [35]. The preparation (0.01 g) was mixed with 10 mL d.$H_2O_2$ and sonicated using an ultrasonic bath (FLAC, LBS2 10LT, Italy) for 20 min. The zeta-potential were measured at a refractive index of 1.7 and an absorbance of 0.01 at room temperature (23˚C) [36]. A beam of light is directed onto the dispersion of nanoparticles, which scatter the light towards the detector. The PDI indicates the size distribution range of the nanoparticles as well as their stability and uniformity. This measurement indicates the average size of the particles in the sample as well as the correlation between the number of particles of a certain size versus the size of the nanoparticles [37].

The spectra of the preparations (extracts and nanoparticles) from the two cultivars were determined by a Fourier transform infrared spectrophotometer (Shimadzu FTIR-8400 S, Japan); the instrument was equipped with an ATR-8000A accessory. The dried preparations (5 mg) were mixed with 100 mg of spectroscopic-grade potassium bromide (KBr). The mixture was ground using a mortar and pressed using a pellet presser to obtain thin pellets. The pellets were then placed under the IR beam, and the spectra of the preparations were measured in a wavenumber range of 4000–400 $cm^{-1}$ [38].

A scanning electron microscope (SEM-JEOL JSM6360LA, Japan) was used to study the morphology and surface structure of olive leaf preparations. A drop of the nanoparticles' dispersion was spread in a thin layer on a slide and allowed to air-dry. The preparations were then vacuum coated with gold for scanning [39] at a magnification of 10.000 X and an accelerating voltage of 15 kV.

## 2.7 Assessment of bioactive components content

Total phenolic content was assessed with the Folin-Ciocalteu method as follows: 1 mL of Folin-Ciocalteu reagent (0.2 N) was added to 200 μL of each preparation. Then, 800 μL of sodium carbonate (7.5%) was added to the samples, and the mixture was incubated in the dark for 2 h. Subsequently, the absorbance of the samples was measured at 760 nm using a spectro-photometer (Jenway 6405UV/VIS, Stone, Staffordshire, UK); the total phenolics expressed as μg gallic acid/g extract [40, 41].

To determine the flavonoid content, the preparations (1 mL), d.$H_2O$ (4 mL), and sodium nitrite (5%; 300 μL) were mixed and incubated for 5 min; then, 300 μL aluminum chloride (10%) was added to the mixture. The solution was re-incubated for 6 min before adding 2 mL of NaOH (1 mol/mL) and adjusting the volume to 10 mL with d.$H_2O$. Then, the absorbance was measured at 510 nm, and the flavonoid content was expressed as μg/g quercetin in the extract [41, 42].

## 2.8 Biological activity

**2.8.1 DPPH$^\bullet$ scavenging activity.** DPPH scavenging activity was determined according to Do et al. [42]. DPPH$^\bullet$ in methanol (0.2 mM) and preparations were mixed at a ratio of 5:1 v/ v before incubation for 20 min in the dark at room temperature. The absorbance of the samples was measured at 517 nm; the scavenging activity was calculated as shown in Eq (1).

**2.8.2 ABTS$^\bullet$ scavenging activity.** Equal volumes of ABTS$^\bullet$ solution (7 mmol/L) and potassium persulfate (2.4 mmol/L) were mixed and incubated in the dark for 16 h. Then, the reagent was prepared by diluting the mixture with d.$H_2O$ (1:60 v:v) to give an absorbance of 0.701±0.01 at 734 nm. A mixture of 4 mL of the reagent and 10 μL of the preparation was then incubated for 6 min before measuring the absorbance at 734 nm versus the control [43]; the ABTS$^\bullet$ radical scavenging activity was calculated using Eq (1) as follows:

$$\% \text{ Inhibition} = \left[ \left( A_{blank} - A_{sample} \right) / \left( A_{blank} \right) \right] \times 100 \qquad (1)$$

Where: ($A_{blank}$) and ($A_{sample}$) are the absorbance of the control (without preparations) and the samples (containing preparations), respectively.

**2.8.3 Cytotoxicity and anticancer assay.** The cytotoxic and anticancer effects of the extracts were determined on the African green monkey kidney (Vero) and colorectal carci-noma (HCT-116) (American Type Culture Collection (ATCC) cell lines (Manassas, VA, USA). Cells (100 μL) were inoculated at a density of 1 x $10^5$ cells/ mL into a 96-well tissue cul-ture plate (passage number of cells $10^3$); the plates were incubated at 37˚C/24 h to form a com-plete monolayer [44].

The cytotoxicity was carried out through the MTT protocol [45] using OLE, OL/WPNs, and OL/AgNPs at different concentrations of 1000–0 μg/mL in a maintenance medium of RPMI-1640 containing 2% FBS. In addition, the maintenance medium as a negative control, while doxorubicin was used as a positive control. The plates were incubated at 37˚C/24 h. Cells were then examined under an inverted light microscope for physical signs of toxicity, such as partial or complete loss of the monolayer, rounding, shrinkage, or cell granulation, as a routine check step. Then, 20 μL MTT solution (5 mg/ml in PBS) was added to each well, and the plates were shaken at 150 rpm for 5 min and incubated in a 5% $CO_2$ environment at 37˚C/4 h. Then, 200 μL of DMSO was added to each well, and the plates were shaken at 150 rpm for 5 min before reading the optical density (OD) at 560 nm and 620 nm on a microtiter plate reader spectrophotometer (SpectrostarNano, BMG Labtech). After subtracting the OD at 620 from the OD at 560, the resulting OD was directly correlated with cell quantity to calculate the

inhibitory concentration ($IC_{50}$) on the Vero and HCT-116 cells. $IC_{50}$ values were calculated by linear approximation regression [46].

The selectivity index (SI) of each extract was calculated using the following Eq (2) [47]:

$$SI = IC_{50} \text{ on Vero cell line } / IC_{50} \text{ on HCT} - 116 \text{ cell line} \tag{2}$$

## 2.9 Real-time PCR for Cox1 and TNF-α expression

The HCT-116 cell line (as control) and HCT-116 cell lines treated with OLE, OL/Ag-NPs, and OL/WPNs at $IC_{50}$ concentrations were subjected to RNA extraction following the instructions of the RNA extraction kit (Qiagen; Hilden, Germany). Subsequently, the purity and amount of extracted RNA were determined using a UV-Spectrophotometer. Afterward, cDNA synthesis and amplification were performed using an automated reverse transcriptase polymerase chain reaction thermocycler (RT-PCR) (Roto-Gene Q, Qiagen, Hilden, Germany), iScript™ One-Step RT-PCR kit with SYBR® Green Master Mix (Biorad; Hercules, California, United States), and primers for Cox1 and TNF-α (Table 1). cDNA was synthesized by Moloney Murine Leukemia Virus (MMLV) reverse transcriptase at 50˚C for 10 min, followed by 5 min at 95˚C to inactivate the enzyme. The amplification was performed through 45 cycles (10 sec each at 95˚C), followed by a 30-sec cycle at 55–60˚C. Expression of the housekeeping gene GAPDH was used to normalize the results [48]. The $2^{\wedge (-\Delta\Delta Ct)}$ method was used to determine the gene expression change; results are presented as fold change compared to the control group [49].

## 2.10 Statistical analysis

Data were analyzed using one-way ANOVA and independent-samples t-test with IBM SPSS 25 (Armonk, New York, United States); the obtained data were expressed as mean ± standard deviation (SD). Differences between means from one-way ANOVA were compared using Duncan's test at a 95% confidence level ($p < 0.05$).

# 3. Results and discussion

## 3.1. Characterization of the silver-nanoparticles and encapsulated particles

**3.1.1. Particle size, Ƹ-Potential, and PDI.** Table 2 shows the particle size, Ƹ-potential, and PDI of nano-silver and WPN-encapsulated particles. Nano-silver technology resulted in smaller particles than WPN-encapsulation: the size of OL/Ag-NPs was 6–12 times smaller than that of OL/WPNs; the particle size of OL/WPNs (Shemlali) was almost twice that of OL/WPNs (Tofahy) ($p < 0.05$). The obtained result was consistent with the phenolic content results (Fig 1a), as Shemlali contained more phenols than Tofahy. Consequently, the concentration of WPI was insufficient to encapsulate all the phenols in the Shemlali extract, resulting in larger particle sizes [50]. Similarly, Soleimanifar et al. [50] obtained a particle size of 232.3–659.8 nm when they encapsulated olive leaf extract with different concentrations of WPC (15–30%), as the particle size was reduced with increasing WPC concentration. Similarly, Akcicek et al. [51] reported that encapsulation of olive pomace extract with chia and rocket gums resulted in particle sizes of 312 nm and 490 nm, respectively. On the other hand, the present study obtained smaller nano-silver particles than those reported by Halob et al. [52] (80.60–126.54 nm) and Rashidipour & Heydari [53] (90 nm).

PDI reflects particle homogeneity: a low PDI indicates homogeneity of particle size [50]. Accordingly, OL/Ag-NPs of Tofahy and Shemlali (PDI = 0.39) are more homogeneous than OL/WPNs of Tofahy (PDI = 0.52), followed by OL/WPNs of Shemlali (PDI = 0.72). The PDI

**Table 1. Primers sequence for RT-PCR.**

| Gene | Forward | Reverse |
|---|---|---|
| TNF-α | 5'–ATGTTTTCTGACGGCAACTTC –3' | 5'–AGTCCAATGTCCAGCCCAT –3' |
| CYC | 5'–AAGGGAGGCAAGCACAAGACTG –3' | 5'–CTCCATCAGTGTATCCTCTCCC –3' |
| GAPDH | 5'– GTCTCCTCTGACTTCAACAGCG–3' | 5'– ACCACCCTGTTGCTGTAGCCAA–3' |

**TNF- α)**: *tumor necrosis factor α*; **(CYC)**: *cytochrome C*; **(GAPDH)**: glyceraldehyde-3-phosphate dehydrogenase

of the encapsulated particles in the present study is consistent with the PDI of olive pomace encapsulated with chia gum (0.514) and rocket gum (0.483) [51]. However, it was higher than the PDI of olive leaf extract encapsulated with different concentrations of WPC (0.074–0.650) reported by Soleimanifar et al. [50].

The ⴹ-potential reflects the stability of NPs in dispersion, as a high ⴹ-potential reflects high electrostatic repulsion, low particle aggregation, and high stability. Besides, the high charges on the particles' surface enable effective interaction with the cells and enhance the delivery of phenols [50]. ⴹ-potential greater than +25 mV or less than -25 mV usually exhibits a high degree of stability, as particles with low ⴹ-potential tend to aggregate due to the attractive force of the particles [53]. Generally, nano-silver and encapsulated olive leaf particles have negative charges due to the negative carboxyl groups of olive leaves [50]. OL/Ag-NPs share a higher ⴹ-potential than OL/WPNs particles. The OL/Ag-NPs of Shemlali were the highest (-31.76 ± 0.87 mV), while the OL/WPNs particles of Shemlali were the lowest (-4.77 ± 0.54 mV). OL/Ag-NPs' ⴹ-potential ranged from -21 mV to -31 mV, consistent with the -25.3 mV determined by Rashidipour & Heydari [53] and the -17.78 to -23.66 mV reported by Halob et al. [52]. On the other hand, the ⴹ-Potential of OL/WPNs was -4.77 mV to -9.74 mV, which is inconsistent with the previous literature. Soleimanifar et al. [50] reported that olive leaf extract encapsulated with WPC had ⴹ-potential of -64.7 and -67.5 mV; Akcicek et al. [51] detected ⴹ-potential values of -29.9 mV and -22.6 mV for olive pomace encapsulated with chia gum and rocket gum, respectively.

**3.1.2 Morphology of the particles.** The diversity of nanoparticle fabrication methods results in particles with various morphological properties (shape, size, and surface). Therefore,

**Table 2. Characterization of silver nanoparticles reduced by olive leaf water extract (OL/Ag-NPs) and olive leaf extract encapsulated by whey protein isolate (OL/WPNs).**

| | OL/Ag-NPs | OL/WPNs |
|---|---|---|
| **Particle size (nm)** | | |
| Tofahy | 37.46 ± 1.85 [bA] | 227.20 ± 2.43 [aB] |
| Shemlali | 44.86 ± 1.62 [bA] | 553.02 ± 3.60 [aA] |
| **ⴹ-Potential (mV)** | | |
| Tofahy | -21.50 ± 1.80 [aB] | -9.74 ± 0.65 [bA] |
| Shemlali | -31.76 ± 0.87 [aA] | -4.77 ± 0.54 [bB] |
| **polydispersity index** | | |
| Tofahy | 0.39 ± 0.08 [aA] | 0.52 ± 0.03 [aB] |
| Shemlali | 0.39 ± 0.03 [bA] | 0.72 ± 0.02 [aA] |

The values are means ± SD of triplicate analysis.

Values with different capital letters (A, B) within the same column indicate a significant difference between Shemlali and Tofahy ($p < 0.05$); Values with different small letters (a-c) within the same row indicate significant differences among different preparations ($p < 0.05$).

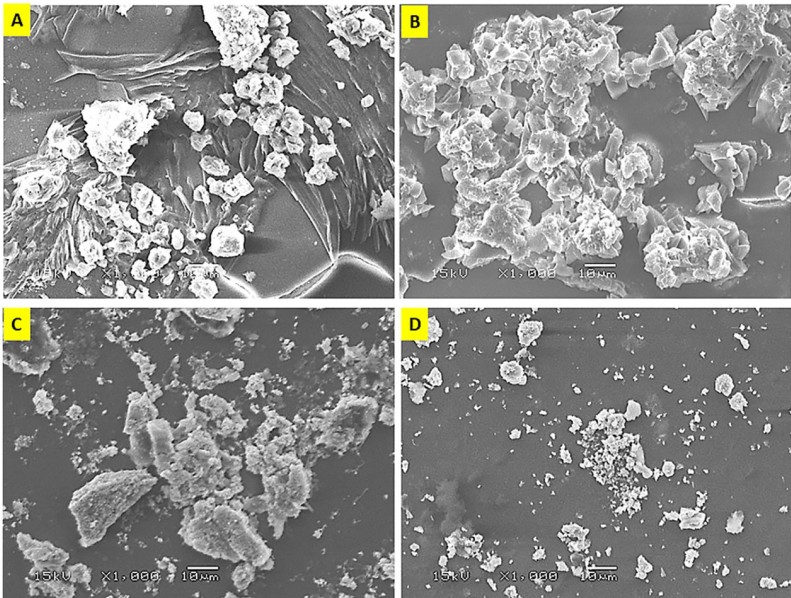

**Fig 1. SEM analysis of encapsulated olive leaf nanoparticles (OL/WPNs) of Tofahy (A) and Shemlali (B); SEM analysis of Ag-NPs reduced by olive leaf extracts (OL/Ag-NPs) of Tofahy (C) and Shemlali (D) varieties.**

SEM is one of the authentic methods to characterize the morphology of the fabricated NPS. Although both encapsulation (Fig 1A and 1B) and nano-silver (Fig 1C and 1D) resulted in irregular NPs, the nano-silver technology yielded smaller particles with a lower aggregation tendency than encapsulation.

Different wall materials, encapsulation methodologies, and pH values affect the size and shape of the particles. Accordingly, the obtained particles' shapes in the present study (Fig 1) were inconsistent with previous literature [50, 54]. Besides, SEM showed the presence of voids on the particles' surfaces, which could be due to shrinkage after drying the biopolymer wall [50]. Shemlali OL/WPNs exhibited higher aggregation tendency than Tofahy OL/WPNs (Fig 1A and 1B), which could be interpreted as Shemlali OL/WPNs having larger particle sizes with a lower ℰ-potential value than Tofahy OL/WPNs (Table 2).

On the other hand, OL/Ag-NPs tended to be quasi-spherical particles, agreeing with the results of Khalil et al. [55] and Nasir et al. [22]. Shemlali OL/Ag-NPs exhibited the least aggregation, with small and homogeneous particles (Fig 1D). The obtained result was in line with the highest ℰ-potential (-31.76 mV) and lowest PDI values (0.39) of OL/Ag-NPs, denoting the low tendency of the particles to aggregate. Tofahy OL/Ag-NPs showed slight aggregation (Fig 1C) due to their ℰ-potential value of -21.50 mV (Table 2), which partially increased their particle size. Moreover, the higher phenolics of Shemlali (OLE) (Fig 3A) reduced and capped the Ag$^+$ in Shemlali OL/Ag-NPs more effectively than in Tofahy OL/Ag-NPs. Accordingly, Shemlali OL/Ag-NPs had fewer agglomerates and more homogeneous particles.

**3.1.3 Fourier transform infrared spectroscopy (FTIR).** FTIR demonstrates the functional groups of bioactive components. Therefore, FTIR shows the possible interaction between the bioactive components and the materials used to fabricate the NPs [30]. Fig 2 illustrates the FTIR of Tofahy and Shemlali (OLE, OL/WPNs, and OL/Ag-NPs): We found similarity in the composition of Tofahy and Shemlali OLEs (S1 Fig); the effectiveness of

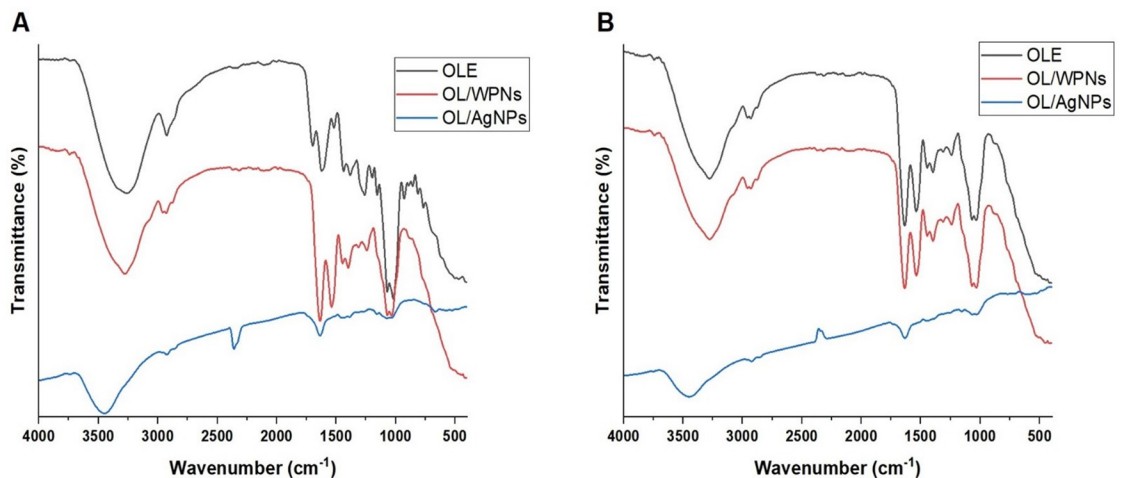

**Fig 2. FTIR analysis of different olive leaf extract preparations of Tofahy (A) and Shemlali (B) varieties Olive leaf extract (OLE), olive leaf extract encapsulated by whey protein isolate nanoparticles (OL/WPNs), and silver nanoparticles of olive leaf extract (OL/Ag-NPs).**

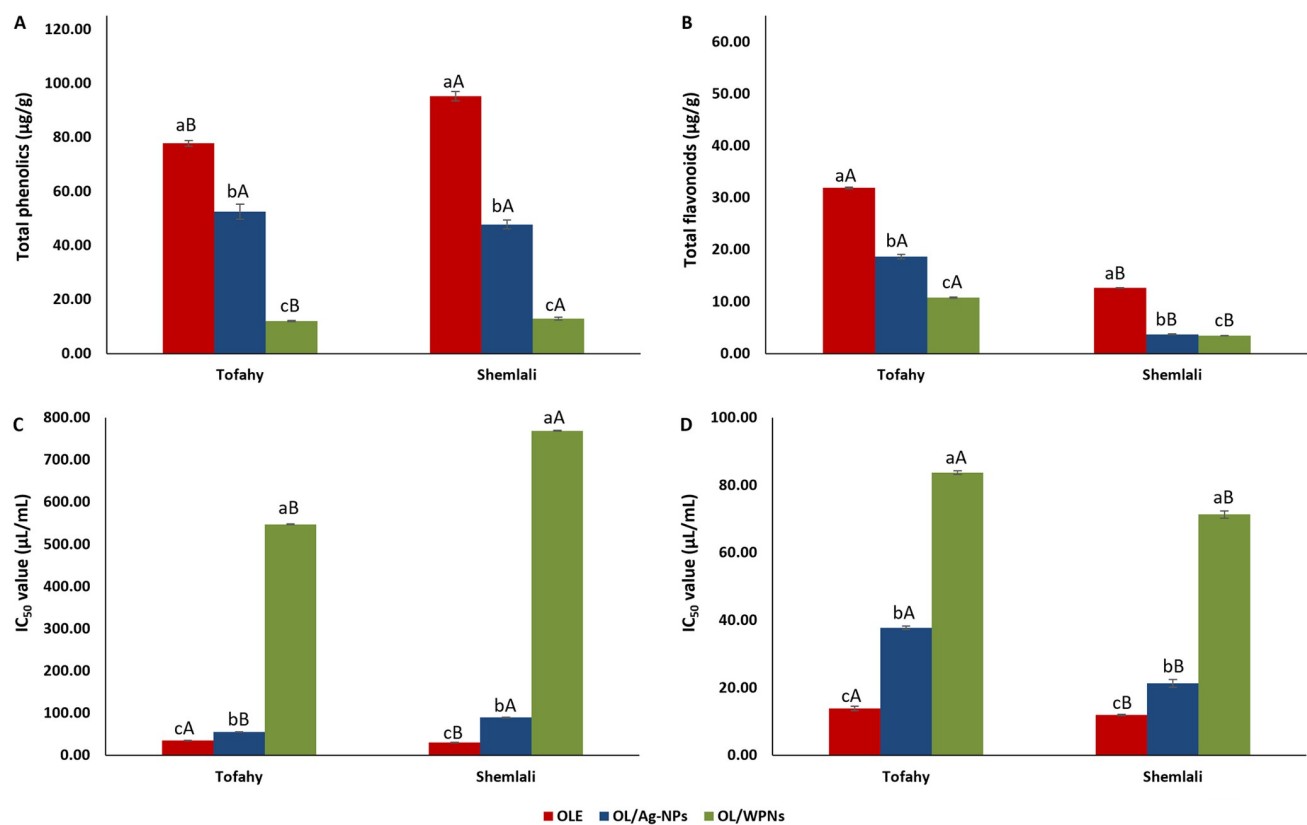

**Fig 3. Total phenolics content (mg/g) (A), total flavonoids content (mg/g) (B), IC$_{50}$ of DPPH$^{\bullet}$ (µL/mL) (C), and IC$_{50}$ of ABTS$^{\bullet}$ (µL/mL) (D) of different extract's preparations from Tofahy and Shemlali varieties: Olive leaf extract (OLE), silver nanoparticles of olive leaf extract (OL/Ag-NPs), and olive leaf extract encapsulated by whey protein isolate nanoparticles (OL/WPNs).** The values are means ± SD of triplicate analysis. Values with different small letters (a-c) within the same variety indicate significant differences among different preparations ($p < 0.05$). Values with different capital letters (A, B) indicate a significant difference between the same extract preparation in Shemlali and Tofahy ($p < 0.05$).

encapsulating OLE with WPNs in OL/WPNs (S2 Fig); the ability of OLE to reduce and cap Ag molecules in OL/Ag-NPs (S3 Fig).

In Tofahy (Fig 2A; S1 Fig) and Shemlali (Fig 2B; S1 Fig) OLEs, we noticed O–H stretching overlapping with N–H at wavenumbers of 3276.97 and 3256.30 cm$^{-1}$ and C = O stretching of a carboxylic acid at 1697.50 and 1695.63 cm$^{-1}$ in Tofahy and Shemlali OLEs, respectively [56]. These vibrations were accompanied by C–O stretching at 1260.18 and 1262.00 cm$^{-1}$ and OH bending at 927.86 and 922.48 cm$^{-1}$ in Tofahy and Shemlali, respectively. We also observed stretching for C = C in the aromatic ring of the phenolic compounds at 1624.46 cm$^{-1}$ (Tofahy) and 1600.98 cm$^{-1}$ (Shimlali) and a glycosidic C–O group in oleuropein at 1069.94 cm$^{-1}$ (Tofahy) and 1069.52 cm$^{-1}$ (Shimlali) [57]. These functional groups reflect that oleuropein, apigenin-7-glucoside, and luteolin-7-glucoside are the phytochemicals in olive leaf extracts [58]. We also observed symmetric stretching for a methylene group (CH$_3$) at 2925.45 (Tofahy) and 2916.65 cm$^{-1}$ (Shemlali), followed by CH scissoring at 1437.88 cm$^{-1}$ in Tofahy, CH rocking at 1381.97 (Tofahy) and 1386.65 cm$^{-1}$ (Shemlali), and in-plane bending vibrations of CH$_2$ at 766.11 and 764.63 cm$^{-1}$ in Tofahy and Shemlali, respectively. These stretching and bending vibrations indicate the presence of an alkane. We also noticed amide II or N–H stretching at 1518.81 and 1521.75 cm$^{-1}$ and C–N at 1018.58 and 1021.34 cm$^{-1}$ due to vibrations of amines [59]. Accordingly, OLEs contain organic functional groups such as alkanes, aromatic compounds, and amide linkages of protein and amine [58], which contribute to the efficient evolution of the NPs.

Encapsulation of OLE with WPNs (OL/WPNs) shifted the peaks and changed their intensity; some peaks disappeared (S2 Fig). The obtained result is consistent with Shakoury et al. [30], who reported that the binding between the loaded material and the WPI restricts the bending and stretching vibrations of the functional groups and disappears the spectrum bands. Besides, the heat treatment during the encapsulation shifted the spectrum bands due to impairing the molecular bonding of the WPI [50]. Encapsulation slightly shifted the O–H and N–H stretching to 3276.92 cm$^{-1}$ and 3275.45 cm$^{-1}$ in Tofahy and Shemlali OL/WPNs, respectively. A new stretching vibration was recorded at 3743.68 cm$^{-1}$ and 3743.05 cm$^{-1}$; this could represent free O–H in alcohol. The C–H stretching for the alkane was changed to an asymmetric stretching in Tofahy OL/WPNs (2958.39–2929.48 cm$^{-1}$) and Shemlali OL/WPNs (2958.23–2930.09 cm). This shift was accompanied by a shift of the C–H scissoring and rocking vibrations to 1445.58–1397.44 cm$^{-1}$ (Tofahy) and 1394.68–1397.44 cm$^{-1}$ (Shemlali). Moreover, the C = O stretching vanished, indicating the connections between the carboxyl and carbonyl groups of the olive polyphenols and the WPI biopolymer [54]. Besides, FTIR revealed a shift in the C = C stretching to 1634.83 cm$^{-1}$ and 1635.05 cm$^{-1}$. Also, amide II or N–H stretching showed a shift of 16 and 19 cm$^{-1}$ in Tofahy and Shemlali OL/WPNs compared to OLEs. In the fingerprint region, we found C–O stretching (1312.99 and 1315.56 cm$^{-1}$), O–H bending (1242.03 and 1241.50 cm$^{-1}$) and stretching (1069.56 and 1069.05 cm$^{-1}$) for the glycosidic C–O group in oleuropein and the C–O–H bending of hydroxyl functions. These shifts indicate the formation of a strong bond between oleuropein and protein [57].

Similarly, FTIR analysis of OL/Ag-NPs proved the ability of the olive leaf extracts to reduce the Ag$^{+1}$ to Ag$^0$ and stabilize the nano biomolecules (S3 Fig). Generally, we found a shift in the vibrations with a remarkable change in intensity; we noticed that some peaks vanished due to the binding between the functional groups of the phytochemicals in OLE and the Ag ions [58]. The stretching vibrations of O–H and N–H are largely shifted to 3449.76 cm$^{-1}$ and 3448.48 cm$^{-1}$ in Tofahy and Shemlali, respectively. Likewise, an O–H stretching for free alcohol was detected at 3740.59 cm$^{-1}$ in Tofahy OL/Ag-NPs. Besides, CH3 stretching in Shemlali was converted to an asymmetric stretching (2923.79–2857.01 cm$^{-1}$), while in Tofahy, it shifted to 2923.93 cm$^{-1}$, with a lower intensity than OLE. Moreover, the C = O stretching shifted to

2359.68 cm$^{-1}$ and 2285.73 cm$^{-1}$ with a lower intensity. The significant shift of O–H and C = O stretching vibrations indicates their binding with silver ions [55, 60]. Besides, amide II or N-H vibrations shifted to 1636.03 cm$^{-1}$ (Tofahy) and 1634.15 cm$^{-1}$ (Shemlali), indicating the involvement of the protein functional groups of OLE as capping agents for the Ag ions, stabilizing Ag-NPs and preventing their agglomeration [22, 61]. Moreover, a large change in the intensity was noticed in the stretching vibration of polysaccharides at 1068 cm$^{-1}$ (Shemlali) and 1072 cm$^{-1}$ (Tofahy) compared to their intensities in the OLE, which also revealed their participation in the reduction and the stability of the NPs [61].

## 3.2. Bioactive components

Fig 3 and S1 Table show the total phenolics (μg GAE/ g extract) and flavonoids (μg QEE/ g extract) of olive leaf preparations. OLE from Tofahy and Shemlali had the highest phenolics (77.81 ± 1.09 and 95.35 ± 1.73 μg GAE/g extract, respectively) and flavonoids (31.89 ± 0.15 and 12.66 ± 0.06 μg QEE/g extract, respectively). Our result was lower than that of El-Messery et al. [54], who reported a total phenolics and flavonoids of 489.8 mg/100g and 11.4 mg/100g, respectively, and Urzúa et al. [25], who extracted 64.3 mg/g of phenolics from olive leaf extract. Similarly, Soleimanifar et al. [50] extracted higher phenolics (286.46 mg/g) than we obtained in the present study. The variation in the bioactive components could be due to the different cultivars, cultivation methods, solvents, and extraction procedures.

On the other hand, total phenolics (Fig 3A) and flavonoids (Fig 3B) followed the same pattern after forming the nano-silver particles and encapsulated with WPNs, with a higher reduction in OL/WPNs than OL/Ag-NPs compared to OLE extracts ($p < 0.05$). In OL/Ag-NPs extracts, phenolic content was reduced by almost 32.50% and 50% in Tofahy and Shemlali varieties compared to OLE extracts. Similarly, the flavonoid content of Tofahy and Shemlali NPs was reduced by 41.5% and 71%, respectively. The significant reduction of phenolics and flavonoids in OL/Ag-NPs revealed the ability of olive leaf phytochemicals to reduce Ag$^+$ to Ag$^0$ [62]. Likewise, encapsulating olive leaf extract with WPNs resulted in a significant reduction in phenolics content to values of 12.08 ± 0.22 μg /g (Tofahy) and 12.92 ± 0.45 μg /g (Shemlali) and a reduction in flavonoids by 66% and 73% to 10.79 ± 0.13 and 3.47 ± 0.03 μg /g in Tofahy and Shemlali, respectively. The reduction in phenolics and flavonoids after encapsulation with WPNs was similar to that observed when olive leaf extract was encapsulated with whey protein concentrate (WPC) [50] and olive pomace with chia and rocket gums [51], indicating the entrapment of the bioactive components in the wall matrices.

## 3.3 Biological activity

**3.3.1. Antioxidant activity.** The antioxidant activity of the extracts toward DPPH$^{\bullet}$ and ABTS$^{\bullet}$ radicals is shown in Fig 3C and 3D, respectively, and S1 Table. The ability of the extracts to scavenge 50% of the free radicals was as follows in descending order: OLE, OL/Ag-NPs, and OL/WPNs. OLE extracts showed robust antioxidant activity; however, the antioxidant activity of the extracts was reduced at OL/Ag-NPs and OL/WPNs ($p < 0.05$).

Tiny amounts of the OLE revealed a high scavenging activity against DPPH$^{\bullet}$ and ABTS$^{\bullet}$ radicals: Tofahy and Shemlali' OLE extracts effectively scavenged 50% of DPPH$^{\bullet}$ (IC$_{50}$ = 35.44 ± 0.45 and 30.39 ± 0.22 μg/mL, respectively) and ABTS$^{\bullet}$ (IC$_{50}$ = 13.91 ± 0.63 and 11.99 ± 0.22 μg/mL, respectively). The high scavenging activity could be due to the high concentrations of oleuropein, rutin, benzoic acid, salicylic acid, p-hydroxybenzoic acid, and ellagic acid [63]. We obtained a higher scavenging activity in the present study than Urzúa et al. [25], who reported that 0.15 mg/mL scavenged 50% of DPPH$^{\bullet}$.

The scavenging activity was significantly reduced in OL/Ag-NPs and OL/WPNs extracts compared with OLE extracts of Tofahy and Shemlali. In OL/Ag-NPs, $55.53 \pm 0.77$ μg/mL of Tofahy and $89.81 \pm 0.70$ μg/mL of Shemlali scavenged 50% of DPPH$^\bullet$ radicals; $37.78 \pm 0.57$ μg/mL of Tofahy and $21.38 \pm 1.16$ μg/mL of Shemlali scavenged 50% of ABTS$^\bullet$ radicals. The significant reduction in antioxidant activity of silver NPs compared to OLE is attributed to the fact that olive leaf phenolics and flavonoids efficiently reduce $Ag^+$ to $Ag^0$, thus reducing the available phytochemicals to scavenge free radicals [62]. Similarly, a notable reduction in the scavenging activity of Tofahy and Shemlali of OL/WPNs was observed due to the entrapment of the phytochemicals within the WPI, $IC_{50}$ values of $547.42 \pm 0.80$ and $769.32 \pm 1.11$ μg/mL (DPPH$^\bullet$) and $83.67 \pm 0.54$ and $71.34 \pm 1.05$ μg/mL (ABTS$^\bullet$).

**3.3.2. Cytotoxicity and anticancer effect of OLE, OL/Ag-NPs, and OL/En-WPI.** Cancer is a growing and challenging disease; colorectal cancer is one of the most prevalent cancers and ranks third in morbidity. Finding a plant-based anticancer drug is vital due to limited funds and resources in developing countries. A crude extract with an $IC_{50}$ of up to 100 μg/mL could be effective as an anticancer agent [64]. However, if the extract shows an anticancer effect within its safe concentrations, it should be considered an anticancer agent. Because the safety of extracts on normal cells is critical before considering the extract as an anticancer drug, we examined the effect of the extracts on the HCT-116 colorectal cell line versus that of the normal (Vero) cell line.

The viability of Vero and HCT-116 colorectal cell lines were plotted versus different extract concentrations to determine the 50% inhibitory concentrations ($IC_{50}$) of the extracts (Fig 4 and S2 and S3 Tables). The extracts and NPs of Tofahy and Shemlali revealed a safe effect on the Vero cell line (Fig 4A and 4B; S2 Table) and a considerable anticancer effect on the HCT-116 cell line (Fig 4C and 4D; S3 Table).

OLE, OL/Ag-NPs, and OL/WPNs showed safer effects than doxorubicin on Vero cell lines. $IC_{50}$ values of Tofahy and Shemlali OLE, OL/Ag-NPs, and OL/En-WPNs were significantly higher than doxorubicin (151.94–789.25 μg/mL vs. 21.92 μg/mL). Similarly, Rashidipour & Heydari [53] conveyed that olive leaf Ag-NPs were safe for normal cell lines. Besides, the green synthesis of Ag-NPs revealed less genotoxicity than the chemically produced Ag-NPs [65]. However, this is the first study to examine the cytotoxic effect of encapsulated olive leaf extract.

On the other hand, the effect of OLE, OL/Ag-NPs, and OL/WPNs on the colorectal HCT-116 cell line was lower than that of doxorubicin ($p < 0.05$). However, the preparations revealed a comparable activity against colorectal cancer ($IC_{50}$ = 77.54–320.64 μg/mL), which were within their safe concentrations on Vero cells (151.94–789.25 μg/mL) (Table 3). Tofahy OL/WPNs had the highest anticancer effect on HCT-116 cell line ($IC_{50}$ = 77.54 μg/mL), followed by Shemlali OL/Ag-NPs ($IC_{50}$ = 94.58 μg/mL), Tofahy OLE ($IC_{50}$ = 95.86 μg/mL), Tofahy Ag-NPs (IC50 = 100.71 μg/mL), Shemlali OLE (IC50 = 206.87 μg/mL), and Shemlali OL/WPNs (IC50 = 320.64 μg/mL). The anticancer effect of the OLE extracts was consistent with the results of Albogami & Hassan [66], who reported the efficacy of olive leaf water extract in inhibiting the growth of HT-29 cells ($IC_{50}$ = 535.3–198.6 μg/mL after 12–72 h). Nevertheless, the effect of OL/Ag-NPs on HCT-116 was less than that on MCF-7 ($IC_{50}$ = 0.024 μg/mL) [53] and HT-29 ($IC_{50}$ = 5 μg/mL) [60]. However, whey encapsulation of OLE revealed a better cytotoxic effect on HCT-116 than the effect of olive leaf encapsulated with sodium alginate and chitosan on MCF-7 cells [3]. The differences in the results could be due to the different cell types, as the extract induces a phenotypic change during cell differentiation [66].

To obtain a clear overview of the efficacy and safety of the extracts, we calculated the selectivity index (SI) for each extract compared with doxorubicin (Table 3). The anticancer agents should strongly select and attack the cancer cells without exerting a cytotoxic effect on the

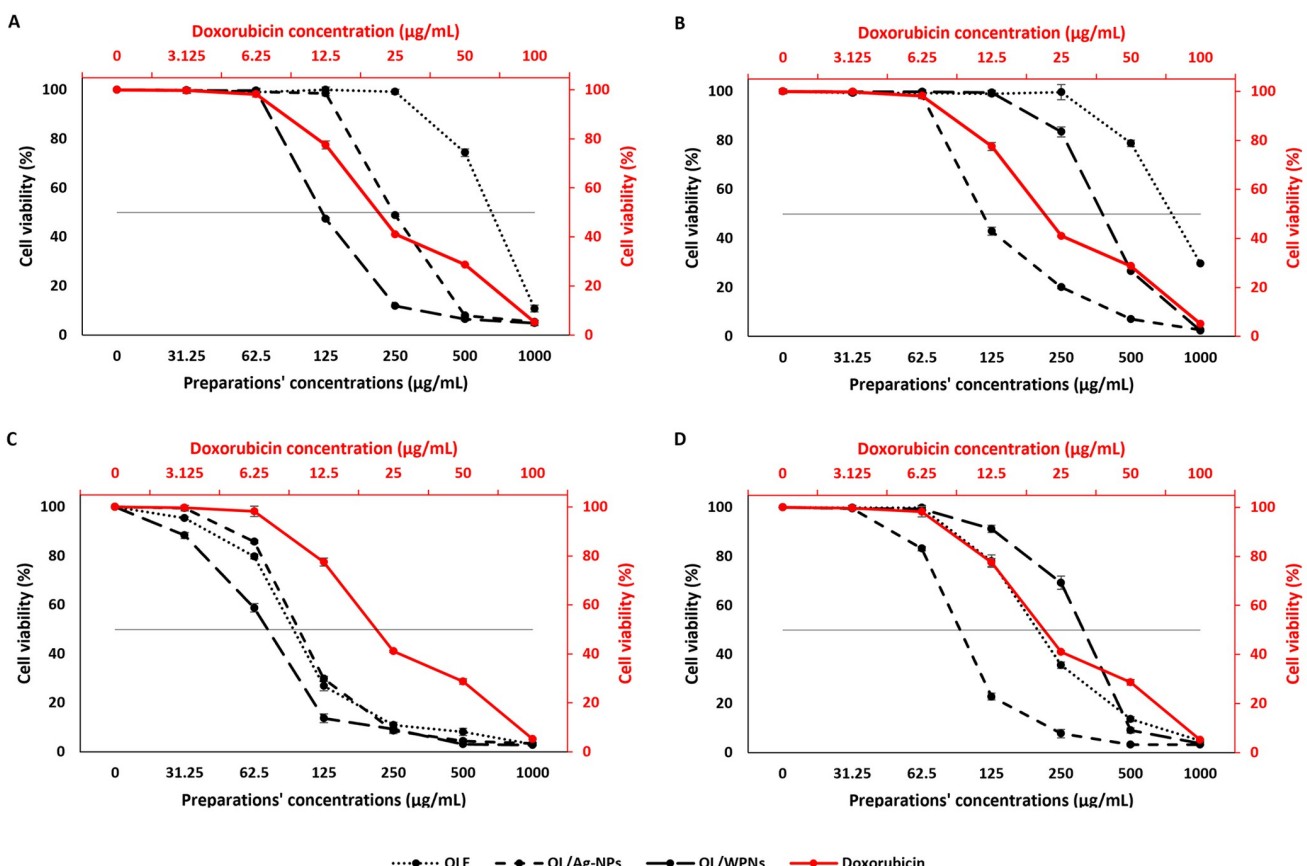

**Fig 4. Cell viability of vero cell line treated with Tofahy (A) and Shemlali extracts (B); cell viability of HCT-116 colorectal cancer cell line treated with Tofahy (C) and Shemlali extracts (D). (OLE)** olive leaf extract, **(OL/Ag-NPs)** silver nanoparticles of olive leaf extract, and **(OL/WPNs)** olive leaf extract encapsulated by whey protein isolate nanoparticles.

normal cells [64]. Therefore, higher SI indicates the higher anticancer activity of the extract [67]. Tofahy OLE, followed by Shemlali OLE, and Tofahy OL/Ag-NPS, had better selectivity (2.97–7.08 μg/mL) than doxorubicin ($P < 0.05$). Otherwise, Tofahy OL/WPNs, followed by Shmlali OL/Ag-NPs and OL/WPNs had weak SI compared to doxorubicin ($P < 0.05$). Therefore, among the prepared NPs, Tofahy OL/Ag-NPs have the best selectivity index, high anticancer activity, and high safety for normal cells.

## 3.4. Gene expression of Cox1 and TNF-α in HCT-116 colorectal cell line

In cancer cells, avoiding apoptosis and cell proliferation promotes cancer metastasis. The cells upregulate the TNF-α expression, which is accompanied by upregulating matrix metalloproteins (MMP2 and MMP9) expression to facilitate cell proliferation [64]. TNF-α controls calcium signal transduction protein in colon cancer cells, and the induction of its upregulation indicates the progression of cancer [68]. Besides, the cells downregulate Cox1 expression by Bcl-2 to prevent cell apoptosis [69]. Therefore, the downregulation of TNF-α expression may inhibit the expression of MMP2 and MMP9 proteins during transcription [70]. In addition, the upregulation of Cox1 may act as a tumor suppressor protein and reduce cell proliferation

**Table 3. IC$_{50}$ and selectivity index (SI) of different preparations of olive leaf extracts from different varieties compared to the reference drug doxorubicin expressed in (μg/mL).**

| | Tofahy | Shemlali | Doxorubicin |
|---|---|---|---|
| **IC$_{50}$ (μg/mL)/ Vero cell line** | | | |
| OLE | 678.87 ± 5.38 [aB] | 789.25 ± 6.56 [aA] | |
| OL/Ag-NPs | 298.96 ± 0.71 [bA] | 153.99 ± 0.75 [cB] | |
| OL/WPNs | 151.94 ± 0.74 [cB] | 390.34 ± 6.97 [bA] | |
| Doxorubicin | | | 21.92 ± 3.16 [*ϙ] |
| **IC$_{50}$ (μg/ml)/ HCT-116 cell line** | | | |
| OLE | 95.86 ± 1.95 [bB] | 206.87 ± 4.31 [bA] | |
| OL/Ag-NPs | 100.71 ± 1.04 [aA] | 94.58 ± 0.77 [cB] | |
| OL/WPNs | 77.54 ± 1.00 [cB] | 320.64 ± 5.02 [aA] | |
| Doxorubicin | | | 9.33 ± 1.70 [*ϙ] |
| **SI (μg/mL)** | | | |
| OLE | 7.08 ± 0.09 [aA] | 3.82 ± 0.06 [aB] | |
| OL/Ag-NPs | 2.97 ± 0.02 [bA] | 1.63 ± 0.01 [bB] | |
| OL/WPNs | 1.96 ± 0.03 [cA] | 1.22 ± 0.01 [cB] | |
| Doxorubicin | | | 2.36 ± 0.14[*ϙ] |

**OLE**: Olive leaf extracts; **OL/Ag-NPs**: silver nanoparticles reduced by olive leaf extracts; and **OL/WPNs** olive leaf extracts encapsulated by whey protein isolate nanoparticles.

The values are means ± SD of triplicate analysis.

Values with different capital letters (A, B) within the same row indicate a significant difference between olive varieties ($p < 0.05$); Values with different small letters (a-c) within the same column indicate significant differences among different extracts' preparations ($p < 0.05$).

[*] indicates a significant difference between doxorubicin and olive extracts' preparations ($p < 0.05$)

[ϙ] indicates a significant difference between doxorubicin and olive varieties ($p < 0.05$)

[69]. In this regard, an effective anticancer drug should prevent oncogenic cell proliferation by regulating gene expression.

Green synthesis of NPs has been shown to have multifunctional biomedical properties, such as apoptosis, inhibition of angiogenesis, reduction of replicative capacity, and targeting of specific signalling molecules and cascades involved in cancer development or progression. In

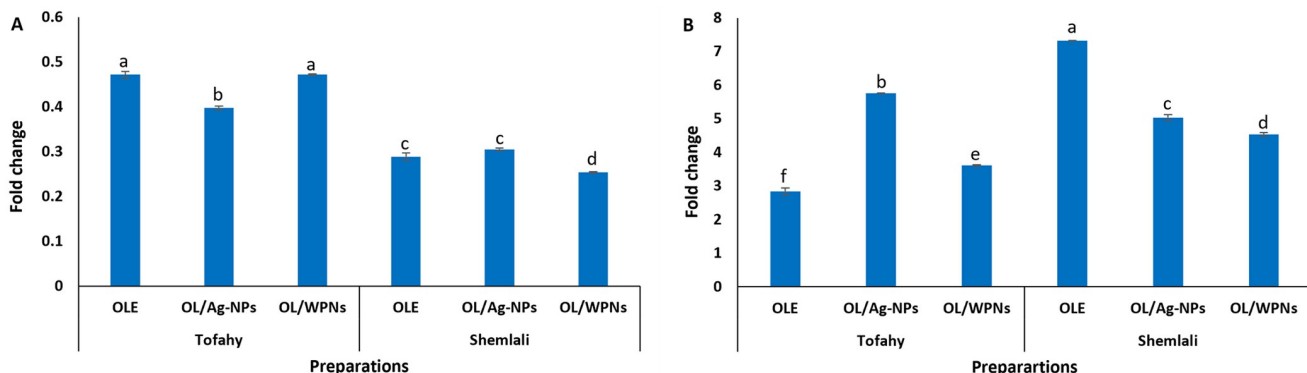

**Fig 5. Expression of tumor necrosis factor α (TNF α) (A) and cytochrome C oxidase (Cox1) (B) in HCT116 cell line treated with different extracts preparations from Tofahy and Shemlali varieties versus that of the control HCT116 cell line (Fold change of 1).** Olive leaf extract **(OLE)**, silver nanoparticles of olive leaf extract **(OL/Ag-NPs)**, and olive leaf extract encapsulated by whey protein isolate nanoparticles **(OL/WPNs)**.

addition, green synthesis of NPs is associated with the inhibition or induction of autophagy in the treatment of cancer, which may increase their ultimate application as biomedicines [71].

Here, Shemlali and Tofahy OLE, OL/WPNs, and OL/Ag-NPs efficiently downregulated TNF-α expression and upregulated Cox1 expression in the treated HCT-116 cell line compared with the untreated cells (control) (Fig 5; S4 Table). Shemlali revealed a higher anti-inflammatory effect than Tofahy due to its higher phenolic content. Shemlali OL/WPNs downregulated the expression of TNF-α by 75% (0.254 ± 0.001), followed by 70% downregulation by Shemlali OLE and OL/Ag-NPs. In comparison, Tofahy downregulated the TNF-α expression by 52% (OLE and OL/WPNs) to 0.472 and 60% (OL/Ag-NPs) to 0.398 ± 0.004 (Fig 5A). Similarly, Ahmed [72] reported that 333 mg/mL olive leaf extract significantly reduced the expression of IL-1b and TNF-α after 2 weeks in patients receiving anticancer chemotherapy.

On the other hand, the extracts superbly upregulated the expression of Cox1 in HCT-116 cells by more than 2.8-fold (p < 0.05). Shemlali was also superior to Tofahy (p < 0.05): OLE, followed by OL/Ag-NPs and OL/WPNs, upregulated the expression of Cox1 to 7.326 ± 0.006, 5.036 ± 0.091, and 4.536 ± 0.051, respectively. In contrast, Tofahy OL/Ag-NPs, followed by OL/WPNs and OLE, upregulated the Cox1 expression to 5.765 ± 0.007, 3.613 ± 0.018, and 2.843 ± 0.103, respectively (Fig 5B; S4 Table). High Cox1 expression indicates cell survival as it suppresses tumor cell growth and induces apoptosis.

Although Tofahy preparations generally exhibited higher cytotoxicity than the Shemlali preparations, the latter showed the most marked regulatory effects on Cox1 and TNF-α. The obtained results suggest that polyphenols from Shemlali preparations promote apoptosis by affecting mitochondrial permeability, downregulating anti-apoptotic genes, upregulating proapoptotic genes, disrupting the mitochondrial membrane, and releasing more Cox1 that activates apoptosis [73]. On the other hand, Tofahy preparations could activate apoptosis through a different pathway, namely the proapoptotic JNK activation pathway.

## 4. Conclusion

Briefly, in olive leaf water extracts of two cultivars (Tofahy and Shemlali), a high amount of phenolics with considerable antioxidant activity was obtained, especially in Shemlali. The production of nano-silver and whey-encapsulated particles from olive leaf water extracts was effectively achieved. Nano-silver technology resulted in smaller and more homogeneous particles with less aggregation tendency than whey-encapsulated particles, particularly from Shemlali. Extracts and NPs were safe for normal cells and revealed considerable anticancer activity on HCT-116 cells with a regulatory effect toward TNF-α and Cox1 expression. They effectively downregulated the TNF-α and upregulated Cox1 expression, notably that of Shemlali. However, Considering the selectivity index, Tofahy OLE, Shemlali OLE, and Tofahy OL/Ag-NPs revealed the best selectivity compared to doxorubicin and other preparations, with a substantial regulatory effect on gene expression.

Accordingly, in the present study, Tofahy OL/Ag-NPs are the best and safest nanoscale-produced particles that can be safely used in food technology. The bioaccessibility, bioavailability, cytotoxicity to various normal cells, anticancer effect on various carcinoma cells, and apoptosis promotion through various genes need to be further explored to get a complete view of the safety and efficiency of the NPs.

## Supporting information

**S1 Fig. FTIR of olive leaf water extract from (A) Tofahy and (B) Shemlali.**
(TIF)

**S2 Fig. FTIR of olive leaf extracts encapsulated by whey protein isolate nanoparticles from (A) Tofahy and (B) Shemlali.**
(TIF)

**S3 Fig. FTIR of silver nanoparticles reduced by olive leaf extracts from (A) Tofahy and (B) Shemlali.**
(TIF)

**S1 Table. Bioactive component and antioxidant activity of olive leaf preparations from two cultivars.**
(DOCX)

**S2 Table. Cytotoxicity of olive leaf preparations from two cultivars versus doxorubicin on Vero cells.**
(DOCX)

**S3 Table. Cytotoxicity of olive leaf preparations from two cultivars versus doxorubicin on HCT-116 cells.**
(DOCX)

**S4 Table. Regulatory effect of olive leaf preparations from two cultivars on the expression of TNF-α and Cox1.**
(DOCX)

## Author Contributions

**Conceptualization:** Hanem M. M. Mansour.

**Data curation:** Eman M. Abdo.

**Formal analysis:** Mohamed G. Shehata, Eman M. Abdo.

**Investigation:** Hanem M. M. Mansour, Mohamed G. Shehata.

**Methodology:** Hanem M. M. Mansour, El-sayed E. Hafez, Amira M. Galal Darwish.

**Resources:** Hanem M. M. Mansour, El-sayed E. Hafez, Amira M. Galal Darwish.

**Validation:** El-sayed E. Hafez, Amira M. Galal Darwish.

**Visualization:** Mohamed G. Shehata, Eman M. Abdo.

**Writing – original draft:** Eman M. Abdo.

**Writing – review & editing:** Mona Mohamad Sharaf, El-sayed E. Hafez, Amira M. Galal Darwish.

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
