## [Decision Letter · Decision Letter 0]

24 Oct 2023

PONE-D-23-32985Comparative Analysis of Silver-Nanoparticles and Whey-Encapsulated Particles from Olive Leaf Water Extracts: Characteristics and Biological ActivityPLOS ONE

Dear Dr. mansour,

Thank you for submitting your manuscript to PLOS ONE. After careful consideration, we feel that it has merit but does not fully meet PLOS ONE’s publication criteria as it currently stands. Therefore, we invite you to submit a revised version of the manuscript that addresses the points raised during the review process.

We look forward to receiving your revised manuscript.

Kind regards,

Marwa Fayed

Academic Editor

PLOS ONE

Additional Editor Comments:

Dear authors

Thanks for your interest in our journal. After careful evaluationfor your manuscript by expert reviewers and my own evaluation we came to the decision that your manuscript still need major revision to be suitable for publication.

Kindly check the reviwers comments and revise your manuscript accordingly.

Best wishes

Reviewers' comments:

Reviewer's Responses to Questions

**Comments to the Author**

1. Is the manuscript technically sound, and do the data support the conclusions?

Reviewer #1: Yes

Reviewer #2: Yes

Reviewer #3: Yes

Reviewer #4: Partly

2. Has the statistical analysis been performed appropriately and rigorously? 

Reviewer #1: Yes

Reviewer #2: I Don't Know

Reviewer #3: Yes

Reviewer #4: Yes

3. Have the authors made all data underlying the findings in their manuscript fully available?

Reviewer #1: Yes

Reviewer #2: Yes

Reviewer #3: Yes

Reviewer #4: Yes

4. Is the manuscript presented in an intelligible fashion and written in standard English?

Reviewer #1: No

Reviewer #2: No

Reviewer #3: Yes

Reviewer #4: Yes

5. Review Comments to the Author

Reviewer #1: This is an interesting study and may be considered for the publication after major revision.

1. There are many grammatical errors that need to be fixed

2. PDI, SEM, FTIR- they need to describe method in details and how it was carried out.

3. Cells (100 μL) were inoculated at a density of 1 X 105 cells/ mL into a 96-well tissue culture plate. What was the passage number of cells that were used in the seeding the cells.

4. Figure 4, why control values was not added, please add control value and redraw the figure

5. Figure 1 images are not clear provide better quality images

Reviewer #2: Comments to author

The current work focuses on the “Comparative Analysis of Silver-Nanoparticles and Whey-Encapsulated Particles from Olive Leaf Water Extracts: Characteristics and Biological Activity” is an interesting study, however major issues should be addressed.

1. Manuscript should be carefully revised for typographical, grammatical errors and long sentences that is difficult to interpret like in abstract. .

2. Add some more background information so that the readers can understand, and finally, summarize your main points. Some related references should be cited:

https://www.mdpi.com/1999-4923/14/10/2251

https://link.springer.com/chapter/10.1007/978-981-16-9190-4_10

3, what is this maintenance medium. Do you mean vehicle media? Was vehicle media tested separately?

5. Pls explain why annealing is necessary and why haven’t authors annealed the sample?

6. How did author confirm the formation of crystalline NPs?

7. How was inhibitory concentration (IC50) measured ? What was the method? At least a reference should be given on it.

8. It is mentioned” Cells were then examined for 229 physical signs of toxicity, such as partial or complete loss of the monolayer, rounding, 230 shrinkage, or cell granulation “. How do the authors see these things without any aid?

9 The authors are mentioning Real-time PCR “The HCT-116 cell line (as control) and HCT-116 cell lines treated with OLE, OL/Ag-NPs, 245 and OL/WPNs at nontoxic concentrations were subjected to RNA extraction following the 246 instructions of the RNA extraction kit (Qiagen; Hilden, Germany).” What is the purpose of treatment of cancer cells with nontoxic concentration ? rather it should have been a non cancer cells? Plz explain

10. Results and discussion section. This study has involved number of chemical constituents during the processing of nanoparticles; therefore authors need to discuss and compare the importance of this study with the previously reported green synthesis and other nanoparticles of biomedical applications.

The English of the manuscript should be polished to remove any typographical and grammatical errors.

Reviewer #3: The research executed by the author(s) was good and the results were exhibited well in the manuscript. The segregation of the methodology was written with detailed manner. Even though the author has to revise few comments for the manuscript.

1. In introduction part, write the research gap effectively.

2. Write the significance for the selection of olive plant for this research in- correlation with nanotechnology.

3. Check the complete literature for the work executed earlier by the researchers.

4. The section 2.6 can be merged together.

5. The assessment of phenolics and flavonoids can be written together and given with proper recent references.

6. The cytotoxicity protocol can be minimized and written with reference alone. The detailed explanation is not necessary.

7. Change the sub- heading of section 3 as Results and discussion.

8. The nanoparticle(s) can be denoted as NPs.

9. The line number 312 “SEM of Tofahy and Shemlali OL/WPNs extracts revealed semi-cube particles”- not necessary. The continuation of the SEM discussion can written and then in brackets the figure number can be denoted.

10. In FT- IR the similar results of recent research investigations can be discussed.

11. Check for grammatical errors in few discussion part.

Reviewer #4: Comments to the Author

In this article, the authors have presented the comparative analysis of silver-nanoparticles and whey-encapsulated particles from olive leaf water extracts and presented their characteristics and biological activity. The idea is interesting, however, some key points should be addressed before its publication in as below.

1. The abstract does not have a conclusion.

2. The important findings of this study need to be highlighted in the abstract. What are the innovations of this study and what makes this study different from similar studies in literature? This should be clarified in the introduction.

3. The section of the Introduction is too short. This section should be modified. More information should be added to show the importance of the study. The authors have to explain the green synthesis of metallic nanoparticles. The authors should mention the extracts containing phytochemicals, authors should further discuss the mechanism of secondary metabolites e.g., the formation of AgNPs as an alternative for chemical and physical approaches and plants as eco-friendly methods for the fabrication of metallic nanoparticles and their superiorities over traditional chemical and physical methods. Include a paragraph to explain. The following references should be cited in this section.

https://doi.org/10.3390/pr11020301

https://doi.org/10.3390/molecules27217656

https://doi.org/10.1515/gps-2022-0052

https://doi.org/10.1515/gps-2022-0042

https://doi.org/10.1515/gps-2022-0031

https://doi.org/10.1515/gps-2021-0067

https://doi.org/10.1515/gps-2020-0025

https://doi.org/10.1049/iet-nbt.2018.5133

4. Why were leaves dried in an oven instead of air drying?

5. in section 2.3 it is mentioned “Olive leaves from different cultivars were”. it would be better to mention “Olive leaves from two cultivars were” as only 2 are included in the study.

6. The author has mentioned that Silver nanoparticles were pellet down through centrifugation then washed three times with d.H2O, dried at 50ºC/24 h, and re-dissolved in d.H2O. Were particles dissolved in water? How?

7. The reason behind the selection of the African green monkey kidney (Vero) and colorectal carcinoma (HCT-116) should be justified.

8 The author should emphasize the novelty of the study.

9. The Olive leaves authentication and voucher number information is missing. How were leaves identified?

10 Exploring the functional groups in extract through Fourier Transform Infrared Spectroscopy below references are useful and should be included

a. Trigonella Foenum-Graecum L. Seed Germination Under Sodium Halide Salts Exposure.

b. Biochemical and FT-IR profiling of Tritium aestivum L. seedling in response to sodium fluoride treatment.

11. Figure 1 is not clear. The quality of the Figures should be improved. In Figures 3 and 5, include the error bar.

12. Gene expression of Cox1 and TNF-α was only reported from the HCT-116 colorectal cell line why not from the green monkey kidney (Vero)?

6. PLOS authors have the option to publish the peer review history of their article (what does this mean?). If published, this will include your full peer review and any attached files.

Reviewer #1: No

Reviewer #2: No

Reviewer #3: **Yes: **Dr. V. Manon Mani

Reviewer #4: No

---

## [Author Response · Author response to Decision Letter 0]

24 Nov 2023

Additional Editor Comments:

Dear authors

Thanks for your interest in our journal. After careful evaluation for your manuscript by expert reviewers and my own evaluation we came to the decision that your manuscript still need major revision to be suitable for publication.

Kindly check the reviwers comments and revise your manuscript accordingly.

Best wishes

Dear Dr. Fayed, 

On behalf of the co-authors, I would like to express our gratitude for giving us the opportunity to resubmit our revised manuscript. We appreciate your time and efforts, which have improved the quality of our manuscript. All necessary changes were made based on the comments of the editors/reviewers. The manuscript has been changed accordingly in Track changes, in addition to the unmarked version of the revised paper without Tracked Changes. The detailed responses and corrections are listed below point by point as raised in the editor/reviewer comments.

Financial Disclosure:

The authors would like to draw your attention that: Due to inflation and economic measures in Egypt, the authors are restricted to pay the equivalent of $250 in Egyptian currency per month.

Therefore, the authors would like to urge the editors to consider a special discount due to these circumstances to help Egyptian authors overcome these obstacles related to international publication.

Thank you again for your time and understanding.

Sincerely, 

Eman Abdo 

Assistant professor of Food Technology 

Faculty of Agriculture Saba Basha, Alexandria University

Alexandria, Egypt

Reviewers' comments:

Reviewer's Responses to Questions

Comments to the Author

1. Is the manuscript technically sound, and do the data support the conclusions?

Reviewer #1: Yes

Reviewer #2: Yes

Reviewer #3: Yes

Reviewer #4: Partly

Reply: Thank you for your positive comments

2. Has the statistical analysis been performed appropriately and rigorously?

Reviewer #1: Yes

Reviewer #2: I Don't Know

Reviewer #3: Yes

Reviewer #4: Yes

Reply: Thank you for your positive comments

3. Have the authors made all data underlying the findings in their manuscript fully available?

Reviewer #1: Yes

Reviewer #2: Yes

Reviewer #3: Yes

Reviewer #4: Yes

Reply: Thank you for your positive comments

4. Is the manuscript presented in an intelligible fashion and written in standard English?

Reviewer #1: No

Reviewer #2: No

Reviewer #3: Yes

Reviewer #4: Yes

Reply: Thank you for your positive comments; the manuscript has been carefully revised.

5. Review Comments to the Author

Reviewer #1: 

This is an interesting study and may be considered for the publication after major revision.

Reply: Thank you for your positive comments, which have improved the quality of our manuscript. We appreciate your valuable time and effort.

1. There are many grammatical errors that need to be fixed

Reply: The MS language has been thoroughly revised, and needful amendments have been conducted. Thank you.

2. PDI, SEM, FTIR- they need to describe method in details and how it was carried out.

Reply: Thank you for your valuable suggestion. PDI, SEM, and FT-IR procedures have been included within the manuscript.

3. Cells (100 μL) were inoculated at a density of 1 X 105 cells/ mL into a 96-well tissue culture plate. What was the passage number of cells that were used in the seeding the cells. ?

Reply: the passage number of cells that were used in the seeding the cells was 103. 

4. Figure 4, why control values was not added, please add control value and redraw the figure

Reply: Control values were added and figure4 was re-attached. Thank you.

5. Figure 1 images are not clear provide better quality images

Reply: Needful amendments were conducted and figure1 was re-attached. Thank you. 

Reviewer #2: Comments to author

The current work focuses on the “Comparative Analysis of Silver-Nanoparticles and Whey-Encapsulated Particles from Olive Leaf Water Extracts: Characteristics and Biological Activity” is an interesting study, however major issues should be addressed.

Reply: Thank you for your positive comments, which have improved the quality of our manuscript. We appreciate your valuable time and effort.

1. Manuscript should be carefully revised for typographical, grammatical errors and long sentences that is difficult to interpret like in abstract. .

Reply: The MS language was thoroughly revised and needful amendments were conducted to avoid typographical statements and ease interpretation understanding. Thank you.

2. Add some more background information so that the readers can understand, and finally, summarize your main points. Some related references should be cited:

https://www.mdpi.com/1999-4923/14/10/2251

https://link.springer.com/chapter/10.1007/978-981-16-9190-4_10

Reply: Thank you for your insightful suggestion. The introduction has been improved as suggested and the suggested references have been included (references 6 and 15).

3, what is this maintenance medium. Do you mean vehicle media? Was vehicle media tested separately?

Reply: Yes, it could be called vehicle medium. It was tested separately as a negative control, as mentioned in the manuscript.

4. Pls explain why annealing is necessary and why haven’t authors annealed the sample?

Reply: For PCR protocols, annealing is of course required. In the present study, we used a one-step PCR kit in which the annealing step is combined with the extension step. The kit is iScriptTM One-Step RT-PCR Kit with SYBR® Green, Bio-Rad, with the following reaction protocol:

Incubate complete reaction mix in a real-time thermal detection system as follows:

cDNA synthesis: 10 min at 50°C

iScript Reverse transcriptase inactivation: 5 min at 95°C

PCR cycling and detection (30 to 45 cycles): 10 sec at 95°C

 30 sec at 55°C to 60°C (data collection step)

Melt curve analysis (optional): 1 min at 95°C; 1 min at 55°C; 10 sec at 55°C (80 cycles, increasing each by 0.5°C each cycle)

5. How did author confirm the formation of crystalline NPs?

Reply: In the present study, we relied on SEM due to the limited capabilities of the laboratory. In addition, the inverted light microscope showed the clusters of crystalline NPs and confirmed the MTT results.

6. How was inhibitory concentration (IC50) measured ? What was the method? At least a reference should be given on it. 

Reply: Thank you for pointing this out. The inhibitory concentration (IC50) values were calculated by linear approximation regression. The following reference (Hazekawa et al., 2019) was included in the manuscript as reference (34).

Hazekawa M, Nishinakagawa T, Kawakubo‑Yasukochi T, Nakashima M. 2019. Evaluation of IC50 levels immediately after treatment with anticancer reagents using a real‑time cell monitoring device. Experimental and Therapeutic Medicine 18:3197–3205. DOI: 10.3892/etm.2019.7876.

7. It is mentioned” Cells were then examined for physical signs of toxicity, such as partial or complete loss of the monolayer, rounding, shrinkage, or cell granulation “. How do the authors see these things without any aid? 

Reply: Thank you very much for your valuable comment. These morphological parameters are not related to MTT step, but they are considered as a routine check step conducted at the cell line lab under inverted light microscope to confirm their optical observations and findings. 

8. The authors are mentioning Real-time PCR “The HCT-116 cell line (as control) and HCT-116 cell lines treated with OLE, OL/Ag-NPs, and OL/WPNs at nontoxic concentrations were subjected to RNA extraction following the instructions of the RNA extraction kit (Qiagen; Hilden, Germany).” What is the purpose of treatment of cancer cells with nontoxic concentration ? rather it should have been a non cancer cells? Plz explain

Reply: Thank you for pointing this out. You are right, this has been a mistake. 

The sentence was corrected as follows; The HCT-116 cell line (as control) and HCT-116 cell lines treated with OLE, OL/Ag-NPs, and OL/WPNs at IC50 concentrations..”

9. Results and discussion section. This study has involved number of chemical constituents during the processing of nanoparticles; therefore authors need to discuss and compare the importance of this study with the previously reported green synthesis and other nanoparticles of biomedical applications. 

Reply: Thank you for your valuable comment. The discussion has been improved. 

10. The English of the manuscript should be polished to remove any typographical and grammatical errors.

Reply: The MS language was thoroughly revised to be polished and needful amendments were conducted to avoid typographical statements and grammatical errors. Thank you. 

Reviewer #3:

The research executed by the author(s) was good and the results were exhibited well in the manuscript. The segregation of the methodology was written with detailed manner. 

Reply: Thank you for your positive comments that truly enhanced the quality of our manuscript. We appreciate your valuable time and efforts.

Even though the author has to revise few comments for the manuscript.

1. In introduction part, write the research gap effectively.

Reply: Thank you for your valuable comment. The gap was written as follows in the introduction section:

“The previous studies focused on the characterization of the prepared nano-silver and encapsulated nanoparticles from olive leaves and their potential anticancer effects, but which particle could successfully and effectively deliver the most phenols to have a potent anticancer effect. To our knowledge, there are no previous studies that have compared the effects of nano-silver and encapsulation technologies on the properties and safety of olive leaves particles and their anti-colorectal activity through the regulation of cytochrome C oxidase (Cox1) and tumor necrosis factor-α (TNF- α) expression.”

2. Write the significance for the selection of olive plant for this research in- correlation with nanotechnology.

Reply: Thank you for your thoughtful suggestion. This part was added to the introduction as follows:

“Therefore, micro and nanotechnologies, such as silver nanotechnology and encapsulation are used to overcome these limitations. Nanoparticles (NPs) can be engineered to encapsulate and protect phytochemicals from degradation, improve their solubility, and increase their bioavailability. In addition, NPs can be functionalized with targeted ligands to selectively deliver drugs to specific cells or tissues to reduce off-target effects and improve therapeutic efficacy. NPs can also be designed to release olive phytochemicals in a controlled manner, enabling sustained drug release over a long period of time (6).”

3. Check the complete literature for the work executed earlier by the researchers.

Reply: Thank you for your insightful comment. The literature has been checked and updated according to the latest studies.

4. The section 2.6 can be merged together.

Reply: Section 2.6 was merged and modified accordingly. Thank you for your valuable suggestion. 

5. The assessment of phenolics and flavonoids can be written together and given with proper recent references.

Reply: Section 2.7 was merged, and needful modifications were conducted.

6. The cytotoxicity protocol can be minimized and written with reference alone. The detailed explanation is not necessary.

Reply: Needful amendments were conducted with respect to other requirements by other respectful reviewers.

7. Change the sub- heading of section 3 as Results and discussion.

Reply: Thank you for pointing this out. The heading has been corrected.

8. The nanoparticle(s) can be denoted as NPs.

Reply: Thank you for your valuable comment. The term “nanoparticle(s)” has been abbreviated as NPs throughout the manuscript.

9. The line number 312 “SEM of Tofahy and Shemlali OL/WPNs extracts revealed semi-cube particles”- not necessary. The continuation of the SEM discussion can written and then in brackets the figure number can be denoted.

Reply: Thank you for the valuable suggestion; the sentence has been updated according to your valuable advice.

10. In FT- IR the similar results of recent research investigations can be discussed.

Reply: Thank you for your valuable comment. The FT-IR discussion has been updated.

11. Check for grammatical errors in few discussion part.

Reply: The MS language was thoroughly revised to be polished and needful amendments were conducted to avoid grammatical errors. Thank you.

Reviewer #4: Comments to the Author

In this article, the authors have presented the comparative analysis of silver-nanoparticles and whey-encapsulated particles from olive leaf water extracts and presented their characteristics and biological activity. The idea is interesting, however, some key points should be addressed before its publication in as below.

Reply: Thank you for your positive comments, which have improved the quality of our manuscript. We appreciate your valuable time and effort.

1. The abstract does not have a conclusion.

Reply: Thank you for pointing this out. A brief conclusion has been added to the abstract.

“(OL/Ag-NPs) from Tofahy and Shemlali showed to be more stable, effective, and safe than (OL/WPNs). Consequently, OL/Ag-NPs, especially Tofahy, are the best and safest nanoscale particles that can be safely used in food and pharmaceutical applications.”

2. The important findings of this study need to be highlighted in the abstract. What are the innovations of this study and what makes this study different from similar studies in literature? This should be clarified in the introduction.

Reply: Thank you for your valuable comment. The abstract has been updated to include the recommended details.

3. The section of the Introduction is too short. This section should be modified. More information should be added to show the importance of the study. The authors have to explain the green synthesis of metallic nanoparticles. The authors should mention the extracts containing phytochemicals, authors should further discuss the mechanism of secondary metabolites e.g., the formation of AgNPs as an alternative for chemical and physical approaches and plants as eco-friendly methods for the fabrication of metallic nanoparticles and their superiorities over traditional chemical and physical methods. Include a paragraph to explain. The following references should be cited in this section.

https://doi.org/10.3390/pr11020301

https://doi.org/10.3390/molecules27217656

https://doi.org/10.1515/gps-2022-0052

https://doi.org/10.1515/gps-2022-0042

https://doi.org/10.1515/gps-2022-0031

https://doi.org/10.1515/gps-2021-0067

https://doi.org/10.1515/gps-2020-0025

https://doi.org/10.1049/iet-nbt.2018.5133

Reply: Thank you for your insightful suggestion. A new paragraph has been added to the introduction explaining the efficiency of green synthesis according to the recommended references.

“Green synthesis of silver nanoparticles (AgNPs) is an innovative, eco-friendly, and sustainable alternative to traditional chemical and physical synthesis methods. This approach utilizing plant extracts due to their high content of phenolic acids, flavonoids, and amides that can reduce and cap silver ions to form stable silver nanoparticles (7-9); thus, providing an environmentally friendly, less toxic, and cost-efficient nanoparticles (9, 10). These green-synthesized nanoparticles have shown promising applications in antibacterial (7, 8, 11) and anticancer therapies (12-14), highlighting their superiority over nanoparticles produced by conventional methods.”

4. Why were leaves dried in an oven instead of air drying?

Reply: Seeking deceased time consuming and contamination for research purposes, however, air drying is more eco-friendly that should be considered in future work. Thank you.

5. in section 2.3 it is mentioned “Olive leaves from different cultivars were”. it would be better to mention “Olive leaves from two cultivars were” as only 2 are included in the study.

Reply: Thank you for your valuable comment. The sentence has been corrected as recommended to be as follows: “Olive leaves of the two cultivars..”

6. The author has mentioned that Silver nanoparticles were pellet down through centrifugation then washed three times with d.H2O, dried at 50ºC/24 h, and re-dissolved in d.H2O. Were particles dissolved in water? How?

Reply: Yes, it dissolved, but it took longer time than usual. It took around 30 min to dissolve completely using magnetic stirrer. This could be because the extracted phenolics were originally the ones that dissolved in water during the extraction. Thank you for your comment.

7. The reason behind the selection of the African green monkey kidney (Vero) and colorectal carcinoma (HCT-116) should be justified.

Reply: Cell lines from monkeys have been widely used to determine cytotoxicity because of their similar structure and immune response to human cells. Vijayarathna and Sasidharan (2012) compared the effect of Elaeis guineensis extracts on Vero cells with that of MCF -7. Badisa et al. (2003) also compared human cancer cell lines (Caco-2, HePG-2, and MCF -7) with mouse adipose arelar cell line. In addition, Mfotie et al. (2017) evaluated the cytotoxicity of extracts from Sarcocephalus pobeguinii against MCF -7, HeLa, Caco-2, and A549 cells compared to Vero cells as normal cells. However, we will consider using normal human cell lines in our future experiments.

1- Vijayarathna, S., & Sasidharan, S. (2012). Cytotoxicity of methanol extracts of Elaeis guineensis on MCF-7 and Vero cell lines. Asian pacific journal of tropical biomedicine, 2(10), 826-829.

2- Badisa, R. B., Tzakou, O., Couladis, M., & Pilarinou, E. (2003). Cytotoxic activities of some Greek Labiatae herbs. Phytotherapy research, 17(5), 472-476.

3- Mfotie Njoya, E., Munvera, A. M., Mkounga, P., Nkengfack, A. E., & McGaw, L. J. (2017). Phytochemical analysis with free radical scavenging, nitric oxide inhibition and antiproliferative activity of Sarcocephalus pobeguinii extracts. BMC complementary and alternative medicine, 17, 1-9.

8 The author should emphasize the novelty of the study.

Reply: Thank you for your valuable comment. The novelty of the study in the introduction as follows:

“The previous studies focused on the characterization of the prepared nano-silver and encapsulated nanoparticles from olive leaves and their potential anticancer effects, but which particle could successfully and effectively deliver the most phenols to have a potent anticancer effect. To our knowledge, there are no previous studies that have compared the effects of nano-silver and encapsulation technologies on the properties and safety of olive leaves particles and their anti-colorectal activity through the regulation of cytochrome C oxidase (Cox1) and tumor necrosis factor-α (TNF- α) expression.”

9. The Olive leaves authentication and voucher number information is missing. How were leaves identified?

Reply: Thank you for your valuable comment. The leaves were were collected from the City of Scientific Research and Technological Applications (SRTA-City) farm, New Burg El-Arab, Alexandria, Egypt and identified by Dr. Yasmin Tawfik, Researcher at Plant Protection and Bio-Molecular Diagnosis Department, Arid Lands Cultivation Research Institute, City of Scientific Research and Technological Applications (SRTA-City), New Borg El Arab, Alexandria, Egypt.

10 Exploring the functional groups in extract through Fourier Transform Infrared Spectroscopy below references are useful and should be included

a. Trigonella Foenum-Graecum L. Seed Germination Under Sodium Halide Salts Exposure.

b. Biochemical and FT-IR profiling of Tritium aestivum L. seedling in response to sodium fluoride treatment.

Reply: Thank you for your valuable comment. The suggested references were included as 52 and 55.

11. Figure 1 is not clear. The quality of the Figures should be improved. In Figures 3 and 5, include the error bar.

Reply: Needful amendments were conducted for figure1, as well as the error bars in Figures 3 and 5 were included and the corrected three figures were re-attached. Thank you.

12. Gene expression of Cox1 and TNF-α was only reported from the HCT-116 colorectal cell line why not from the green monkey kidney (Vero)?

Reply: Due to financial issues; authors preferred the HCT116 colon cancer cell line over Vero for investigations as a sample for gene expression of Cox1 and TNF-α because this cell line was reported to induce high and dose-dependent levels of response to increasing concentrations of genes and can thus be considered to be more sensitive (De Angelis et al., 2004)

De Angelis PM, Kravik KL, Tunheim SH, Haug T, Reichelt WH. 2004. Comparison of gene expression in HCT116 treatment derivatives generated by two different 5-fluorouracil exposure protocols. Molecular Cancer 3:1–11. DOI: 10.1186/1476-4598-3-11.

---

## [Decision Letter · Decision Letter 1]

5 Dec 2023

Comparative Analysis of Silver-Nanoparticles and Whey-Encapsulated Particles from Olive Leaf Water Extracts: Characteristics and Biological Activity

PONE-D-23-32985R1

Dear Dr. Mansor

We’re pleased to inform you that your manuscript has been judged scientifically suitable for publication and will be formally accepted for publication once it meets all outstanding technical requirements.

Kind regards,

Marwa Fayed

Academic Editor

PLOS ONE

Additional Editor Comments (optional):

Thanks for your efforts in revising your manuscript and now it is suitable for publication  

Reviewers' comments:

Reviewer's Responses to Questions

**Comments to the Author**

1. If the authors have adequately addressed your comments raised in a previous round of review and you feel that this manuscript is now acceptable for publication, you may indicate that here to bypass the “Comments to the Author” section, enter your conflict of interest statement in the “Confidential to Editor” section, and submit your "Accept" recommendation.

Reviewer #1: All comments have been addressed

Reviewer #2: All comments have been addressed

Reviewer #4: All comments have been addressed

2. Is the manuscript technically sound, and do the data support the conclusions?

Reviewer #1: Yes

Reviewer #2: Yes

Reviewer #4: Yes

3. Has the statistical analysis been performed appropriately and rigorously? 

Reviewer #1: Yes

Reviewer #2: I Don't Know

Reviewer #4: Yes

4. Have the authors made all data underlying the findings in their manuscript fully available?

Reviewer #1: Yes

Reviewer #2: Yes

Reviewer #4: Yes

5. Is the manuscript presented in an intelligible fashion and written in standard English?

Reviewer #1: Yes

Reviewer #2: Yes

Reviewer #4: Yes

6. Review Comments to the Author

Reviewer #1: The revised manuscript has been improved and all the issues have been addressed in the revised manuscript and may be considered for the publication.

Reviewer #2: All the comments have answered satisfactorily except:

5.The authors are mentioning Real-time PCR “The HCT-116 cell line (as control) and

HCT-116 cell lines treated with OLE, OL/Ag-NPs, and OL/WPNs at nontoxic

concentrations were subjected to RNA extraction following the instructions of the RNA

extraction kit (Qiagen; Hilden, Germany).” What is the purpose of treatment of cancer

cells with nontoxic concentration ? rather it should have been a non cancer cells? Plz

explain:

My question was that authors should have have used non cancerous cell line as control as it is one of the important parameter to take drug further for applicability.

Reviewer #4: Author has addressed all the comments raised by reviewer . Manuscript is acceptable for publication.

7. PLOS authors have the option to publish the peer review history of their article (what does this mean?). If published, this will include your full peer review and any attached files.

Reviewer #1: **Yes: **Firdos Alam Khan

Reviewer #2: **Yes: **Dr Suriya Rehman

Reviewer #4: No

---

## [Editor Report · Acceptance letter]

8 Dec 2023

PONE-D-23-32985R1 

 Comparative Analysis of Silver-Nanoparticles and Whey-Encapsulated Particles from Olive Leaf Water Extracts: Characteristics and Biological Activity

Dear Dr. Mansour:

I'm pleased to inform you that your manuscript has been deemed suitable for publication in PLOS ONE. Congratulations! Your manuscript is now with our production department. 

Kind regards, 

on behalf of

Prof. Marwa Fayed 

Academic Editor

PLOS ONE